# Wellbeing Literacy: A Capability Model for Wellbeing Science and Practice

**DOI:** 10.3390/ijerph18020719

**Published:** 2021-01-15

**Authors:** Lindsay G. Oades, Aaron Jarden, Hanchao Hou, Corina Ozturk, Paige Williams, Gavin R. Slemp, Lanxi Huang

**Affiliations:** Centre for Positive Psychology, Melbourne Graduate School of Education, University of Melbourne, Melbourne, VIC 3010, Australia; lindsay.oades@unimelb.edu.au (L.G.O.); hanchaoh@student.unimelb.edu.au (H.H.); corriely@gmail.com (C.O.); paige@drpaigewilliams.com (P.W.); gavin.slemp@unimelb.edu.au (G.R.S.); lanxi@student.unimelb.edu.au (L.H.)

**Keywords:** wellbeing, literacy, wellbeing literacy, capability

## Abstract

Wellbeing science is the scientific investigation of wellbeing, its’ antecedents and consequences. Alongside growth of wellbeing science is significant interest in wellbeing interventions at individual, organizational and population levels, including measurement of national accounts of wellbeing. In this concept paper, we propose the capability model of wellbeing literacy as a new model for wellbeing science and practice. Wellbeing literacy is defined as a capability to comprehend and compose wellbeing language, across contexts, with the intention of using such language to maintain or improve the wellbeing of oneself, others or the world. Wellbeing literacy is underpinned by a capability model (i.e., what someone is able to be and do), and is based on constructivist (i.e., language shapes reality) and contextualist (i.e., words have different meanings in different contexts) epistemologies. The proposed capability model of wellbeing literacy adds to wellbeing science by providing a tangible way to assess mechanisms learned from wellbeing interventions. Moreover, it provides a framework for practitioners to understand and plan wellbeing communications. Workplaces and families as examples are discussed as relevant contexts for application of wellbeing literacy, and future directions for wellbeing literacy research are outlined.

## 1. Introduction

Wellbeing science is the scientific investigation of wellbeing, its antecedents and consequences. Alongside the growth of wellbeing science has been significant interest in wellbeing interventions at individual, organizational and population levels, including measurement of national accounts of wellbeing. We assert that to achieve population-based wellbeing, a ubiquitous systemic lever is required, and that language use represents such a lever. The central focus of this article addresses how language can be intentionally used to cultivate and sustain wellbeing over time and across contexts. Intentional language use is one mechanism by which individuals learn, define and ultimately influence their own wellbeing. Dominant discourses affect private and public conceptions of wellbeing, leading to potential impacts on individual’s levels of thriving. In this context, the questions ‘how do people communicate about and for wellbeing?’ and ‘why does it matter?’ are central.

Hence, we propose the concept of wellbeing literacy [1], defined as ‘a capability to comprehend and compose wellbeing language, across contexts, with the intention of using such language to maintain or improve the wellbeing of oneself, others or the world’. More broadly, wellbeing literacy relates to how people communicate about and for wellbeing. It is conceptualised as a capability (derived from Sen’s capability approach [2,3,4]) that promotes freedom and choice in the wellbeing experience: Freedom to choose what wellbeing means to a person and choice in how that is maximised via language and knowledge. Wellbeing is highly individual and the freedom and choice to decide what wellbeing means to them, is essential to wellbeing itself. Beyond an idea or construct, wellbeing is also an experience. Language influences experiences [5,6], and as such also wellbeing. In the following section we examine how language and literacy are relevant to wellbeing as they help increase people’s freedom to choose and influence what makes life meaningful for them.

## 2. The Role of Language, Literacy and Context in Wellbeing Literacy

Language provides insights into wellbeing experiences [7]. Over recent years many studies have explored the relationship between language and wellbeing [7,8,9,10,11,12]. The literature outlines at least three perspectives of language (and combinations of these). The first views language as a marker that reflects people’s psychological status and individual differences [7,10], the second views language as a stimulus by which people’s cognition, emotion or behaviour can be passively influenced [5,13] and the third views language as a resource that people use actively to construct their psychological and social realities. Wellbeing literacy is based on this third view of language. 

Language is also ubiquitous and therefore has broad, systemic effects on human social experiences. People actively construct meanings in their experiences through language [14,15,16]. These constructed meanings make language a powerful leverage point for people to influence their own wellbeing and the wellbeing of others. For example, it is through language in conversations that knowledge of wellbeing can be embedded, expanded and co-created—often without conscious intention. As Brothers mentions, “we are in language like a fish is in water. It is only when the fish is taken out of the water, that the person realises that they were in language” [16] (p. 7). 

As a communication system, language is a vehicle for transporting ideas and thoughts between people and processing information within one’s own head [17]. However, language is not just a means of transmitting information; it also serves to actively construct meaning in experiences [14,15,16]. Drawing on a constructivist epistemology and particularly social constructionism, a key assumption of wellbeing literacy is that this meaning making process does not happen in isolation and that language is socially constructed. Social constructionism [17,18] proposed that meanings (and by extension realities) are created by language, often within the interactions between people. As a branch of constructivist epistemology, social constructionists believe that realities are constructed by humans and that there are little (if any) objective realities [17,18]. 

These ideas are consistent with Linguistic Relativity Theory (otherwise known as the Sapir–Whorf Hypotheses [19]), which proposes that the structure of a language influences the language speaker’s cognition and how they view the world. A common, although disputed, example refers to the number of words that Inuit Eskimos have for ‘snow’. There is only one word for snow in the English language, whereas in the Inuit language there are many words for this concept. Therefore, Inuit-speakers can think about and experience snow in ways that non-Inuit-speakers cannot access. Sapir and Whorf argued that people are not often aware of the impact of these linguistic differences until they come across cultures different from their own. 

In addition, studies have shown that individuals can intentionally change their experiences by using language in certain ways, showing that intentional use of language can improve wellbeing of the self (i.e., part of wellbeing literacy related to self). For example, Pennebaker and Seagal found that writing about important personal experiences using more positive emotion words improved mental and physical health [20]. King demonstrated that writing about life goals brought about psychological and physical benefits [21]. Adler found that increases in the theme of agency in a person’s personal narrative preceded improvements in mental health, which implied that individuals tended to live in a way aligned with their narratives [22]. The key point here is that if people can develop language capability related to wellbeing, they have more choices about their construction of wellbeing through language. 

Broadening this idea further, it is important to pay attention to not only one’s own language, but also to the language that is used to generate discussion about important matters in society—otherwise known as dominant discourses. ‘Discourses’ here refers to ways in which social groups think about various aspects of society, and often come from social institutions, such as education, media and politics. Academics in social constructionism [17,18] and contemporary literacy research [23,24,25,26,27] discuss how dominant discourses are embedded in power relations and therefore shape knowledge and relationships in society. As Burr (2003) aptly puts it, “discourses are intimately connected to the institutional and social practices that have a profound impact on how we live our lives, on what we can do and on what can be done to us” [17] (p. 87). For better or for worse, the power of current discourses is more likely to be sustained when there is a lack of awareness of these discourses—that is, when language is not used with intention. With these aspects in mind, it is apt to consider what the current wellbeing discourse is. What are the dominant ideas that prevail about what wellbeing means? Alexandrova asks this question in the context of the philosophy of science of wellbeing and argues that with the recent proliferation of wellbeing investigations comes a risk of ‘wellbeing experts’ imposing rigid, top-down understandings of wellbeing onto the lives of others [28,29]. This is relevant to the issue that there are multiple ways of conceptualising wellbeing, impacted by context. If one accepts a contextualist approach to wellbeing, a single circumscribed definition of wellbeing is not necessary. The study of language use in context is a useful direction to improved understanding of wellbeing.

Discourses arise within contexts and are often only relevant in that specific context. This is consistent with contextualist theories, which argue that behaviours and phenomenon are best understood in their relevant context [30,31]. Context is essential to both understanding and anchoring experiences. Without context, meanings are at best limited, and at worst, entirely lost. In proposing wellbeing literacy as a systemic lever for wellbeing, we challenge the notion that wellbeing is a fixed, universal thing that can be ‘gained’. Instead, we argue that wellbeing is a dynamic evolving process—a collection of experiences that are socially constructed and constantly shifting depending on relevant social context(s). As contexts change across time and space, it becomes essential that information is not learnt and applied too rigidly. Langer speaks to this in her work about the barriers to mindful learning [32]. She states that one of the problems with current learning models is that students are taught to learn the basics of their subject matter, to the point where the skill becomes second nature. She calls this phenomenon ‘overlearning the basics’. The problem with this automatic way of being is that it often becomes a barrier for students when contexts inevitably change, and they struggle to adjust their learnt skills.

This discussion of contextualisation is directly applicable to learning for wellbeing. If ideas about wellbeing are taught and learnt too rigidly, there is the risk of people not knowing what to do when contexts change, or of applying information that is not useful for the new context. Contemporary literacy researchers have arrived at similar ideas about the relevance of context, power and meaning-making to the concept of literacy, with contemporary views of literacy recognising (a) the importance of multiple modes of communication (i.e., literacy is more than reading and writing) [33]; (b) the importance of context [34]; and (c) that language has a user with intentions [35].

When narrowly defined, literacy has traditionally meant the “ability to read and write” [36] (p. 79). However, the concept has evolved over the last five decades [37], largely pushed by scholars and theories under an umbrella of sociocultural views of literacy. These views put significant emphases on the social and cultural contexts in which literacy is practiced, and on the power relations nested in language use [22]. Scholars are now seeing literacy beyond reading and writing as a social practice that operates between individuals, rather than as a cognitive process within individuals. 

The underlying principles of the more recent developments in literacy theory contribute to our work on wellbeing literacy. Keefe and Copeland summarised five core principles of literacy, which we follow in our conceptualising of wellbeing literacy [38]:All people are capable of acquiring literacy.Literacy is a human right and is a fundamental part of the human experience.Literacy is not a trait that resides solely in the individual person. It requires and creates a connection (relationship) with others.Literacy includes communication, contact and the expectation that interaction is possible for all individuals; literacy has the potential to lead to empowerment.Literacy is the collective responsibility of every individual in the community; that is, to develop meaning making with all human modes of communication to transmit and receive information.

The concept of literacy is critical in many areas of life, such as families, communities, workplaces and healthcare settings. As such, its importance has been increasingly noticed by many non-literacy scholars [39] and in various fields, for example, health literacy [40], workplace literacy [41,42] and virtue literacy in education [43,44]. However, literacy’s importance in regard to wellbeing science has yet to be addressed. Given the relevance and importance of literacy with wellbeing, we introduce ‘wellbeing literacy’ below. Different to other literacies which apply literacy in certain external contexts (e.g., health, workplaces, education), wellbeing literacy is related more with people’s inner experience, specifically, the use of language to enlarge people’s capability of experiencing wellbeing and empowering people to practice in ways that improve and sustain theirs and others’ wellbeing.

A number of intellectual and disciplinary fields contribute to the concept of wellbeing literacy as depicted in Figure 1 below. We have explored how wellbeing science, contemporary literacy, constructivism and contextualism are relevant to wellbeing, and we now introduce and discuss the concept of ‘wellbeing literacy’, before presenting the capability model of wellbeing literacy to demonstrate why and how it matters for wellbeing research and practice.

## 3. What Is Wellbeing Literacy?

We define wellbeing literacy as: “the capability of comprehending and composing wellbeing languages, across various contexts, that may be intentionally used to maintain or improve the wellbeing of oneself, others or the world”. By ‘language’ we mean ‘multimodal symbolic systems’, which may be alphabetic, pictorial, visual, aural or combinations of these. The symbolic systems often comprise of vocabulary, grammar and sentence structure, and this understanding of language is aligned with the theory of multiliteracies [45]. By ‘use’ in our understanding of literacy as language use, we mean both comprehending and composing [46]. The reason we do not use reading and writing is because if we accept the multimodal nature of language, use of language is not only reading and writing, but also includes listening, viewing, speaking and creating. Also, by using comprehending and composing, we see people as actively generating and communicating meaning by interacting with texts, as opposed to extracting meaning from texts [14,15]. 

There are five components to this definition, as shown and outlined in more detail in Table 1 below.

First, wellbeing literacy requires vocabulary, knowledge and language skills relevant to wellbeing. Individuals need some proficiency in wellbeing vocabulary (e.g., being able to articulate the things that they value) and wellbeing knowledge (awareness of evidenced-based principles about wellbeing that is relevant to what they value). 

Second, individuals need to be able to comprehend multimodal texts relevant to wellbeing including reading, listening and viewing [46]. In contemporary society, individuals with high wellbeing literacy have access to a range of modalities, for example, wellbeing relevant books, YouTube clips, and blogs. 

Third, wellbeing literacy requires composition of multimodal texts relevant to wellbeing, including writing, creating, and speaking [46]. Similar to comprehension skills, individuals might compose their wellbeing experiences via multiple modalities in a way that is consistent with their values and social context. Examples include verbally expressing their feelings to others, social media posts, writing blogs or singing songs. 

Fourth, we discussed earlier the role of context in wellbeing. By ‘context’, we mean the who, where and when of the language use. Contemporary views frame literacy as a social construct; people develop ‘literacies’ in contexts outside of schools. These other contexts can be physical in nature (e.g., homes, extracurricular activity settings) or digital in nature (e.g., the internet, social media) [47]. There are different ways of communicating depending on the context. For example, in social media, ‘lol’ (meaning ‘laugh out loud’) is an expression unique to instant messaging platforms and makes sense predominately in those contexts; although young people now also verbalise ‘lol’ to each other—an example of literacy evolving. Further, recent perspectives of literacy have led to ‘literacies’ sometimes being used as a metaphor for ‘being competent’. For example, if a person has high emotional literacy [48] they are considered competent in how they understand and use emotions. Health literacy [49,50] and mental health literacy [51,52] are two other examples of the dozens of ‘literacies’ now available. Scholars have thus recognised that literacy is now often context specific. 

People with high wellbeing literacy are aware of differences across contexts and adapt their language to fit the situation in front of them. They recognise that how they communicate to a 12-year-old about and for their wellbeing, is going to be different to how they communicate to a 60-year old, and these would also be differ for each depending on the context (e.g., school vs. beach). The meaning of words, for example, may also vary by context. Hence, the metacognitive skill of adjusting to context is part of being wellbeing literate.

Finally, wellbeing literacy must involve some degree of intentionality—ongoing intention to improve and/or maintain the wellbeing of self, others or the wider world [53]. This aspect involves intention on the part of the person to prioritise wellbeing of the self and/or others. When a person has high wellbeing literacy, they are not only thoughtful of how they are using their language, but they are doing so because they want to improve the wellbeing of self, others or the world. 

A further way to understand intentionality is the idea of being mindful. Although a full analysis of mindfulness is beyond the scope of this paper, by ‘mindful’ we mean being aware of why language is used in certain ways, and intentionally adapting the use of language to meet the needs of certain contexts. Literacy is not only an autonomous and neutral skill [54], it is also a practice with a purpose. Perry emphasises that literacy is “what people do with reading, writing, and texts in real world contexts and why they do it” [55] (p. 54). People intentionally use literacy for specific purposes and they often have multiple functions that may or may not be related to building traditional literacy skills. For example, a person might read a novel to improve their reading skills, for enjoyment, or for relaxation; or a combination of these. Another person might send an SMS to coordinate a meeting time, but also to increase contact with a friend. Literacy can be used for multiple purposes, which supports our definition of wellbeing literacy to include ‘mindful use of language in contexts’.

To summarise, five components interact to create wellbeing literacy as shown in Figure 2 below. 

Importantly, wellbeing literacy is presented as a *capability* [2,3,4] and we examine what this means and its implications for research and practice next.

## 4. Wellbeing Literacy as a Capability

The Capability Approach (CA) [2,3,4] is a needs-based economic theory of wellbeing that highlights the importance of freedom for people to set and choose their own definitions of wellbeing. Pioneered by economist Amyrta Sen [2,3,4] and developed further by philosopher Martha Nussbaum [56,57], the CA proposes that any effective conceptualisation of wellbeing must have genuine opportunities to experience wellbeing, as defined by the person. Sen argued that it is insufficient to measure wellbeing via economic resources or pre-defined outcomes—it is important to factor in what he calls capabilities; ‘what people can be or do’. According to Sen, the societies with the highest wellbeing are the ones that commit to maximising freedom of choice for people. 

‘Functionings’ are capabilities realised—the endpoint of capabilities. Functionings are the things that people seek to ‘be and do’ that are valuable to them, and as such people have multiple functionings in their lives. Sometimes referred to as ‘achievements’, functionings can be elementary in nature (e.g., eating lunch, going to the shops) or more complex (e.g., to love, to be politically aware). According to the CA, the central feature of wellbeing is the opportunity to achieve valuable functionings. Being educated, riding a bike, eating lunch, are all examples of various ways that a person can value to ‘be or do’, in that they are observable expressions of wellbeing. 

A key strength of the Capability Approach is that unlike other economic models, it recognises the limitations of focusing on resources or achievements (functionings) as measures of a society’s wellbeing. For example, Alkire notes that:

“the limits of focusing on achievements for assessing quality of life becomes obvious when considering cases where a low observed functioning (e.g., low calorie intake) reflects a choice (as in the case of fasting), or where a high level of functioning reflects the choice of a benevolent dictator” [58] (p. 5). 

Similarly, the CA recognises that resources are a limited measure of wellbeing as they are only useful if they are able to be ‘converted’ into meaningful wellbeing achievements. In the CA, capabilities (opportunities) allow that conversion to take place. For example, Indira might be given access to a bike (resource), but in order to be able to ride that bike (functioning), Indira would need to have the capability to ride the bike. That capability would be affected by various factors—for example, safe roads to ride on, appropriate attire to ride the bike, and opportunity to safely leave the house. 

The Capability Model of Wellbeing Literacy presents wellbeing literacy as a mediator (or moderator) in the wellbeing experience; a capability (or freedom in terms of a Sen conceptualisation of capability) that influences the experience of wellbeing. Wellbeing literacy allows individuals options of what to *do* with the environmental conditions that are available to them. In these terms, the capability is closely related to the concept of agency [59]. The higher the wellbeing literacy, the more options that are available to the individual to ‘convert’ the wellbeing opportunities in their internal or external environment into meaningful wellbeing achievements. In doing this, they have a better chance of experiencing wellbeing directly, seeing wellbeing of others improve, or support flourishing of the external environment [60]. This is shown in Figure 3 below.

Our model also draws from the Engine of Wellbeing Model [61], which was developed to organise and systematically integrate the array of wellbeing theories in the literature and is organised around three key components: inputs, processes, and outcomes of wellbeing.

The internal environmental conditions that will influence wellbeing literacy, and therefore also wellbeing, may include attributes of a person, for example, genetics, physiology and personality. External environmental conditions may include, but are not limited to, the social environment (e.g., family, friends), economic and educational environments (e.g., employment, national economy, availability of schooling, resources and infrastructure) and physical environments (e.g., clean drinking water, pollution). The original notion of thriving, from *thrifa*, refers to ‘taking hold of the environment’. In this sense, the use of the capability is thriving, as it is taking hold of the internal or external environments, through language use, to achieve a wellbeing experience for self or others, or flourishing of a physical environment.

To provide an example, John wants to experience love/connection with a person (wellbeing experience) because he feels this will be important to his wellbeing. He might be fortunate enough to have access to a community of people (external environmental condition). However, his chances of experiencing love are reduced if he does not have wellbeing literacy capabilities around listening and communicating his needs. He may not have many relationship-building skills (e.g., active listening), and he also may not have access to different modes of communication (e.g., speaking, writing, reading, listening, creating and viewing) that would help him establish connections. Through the development of wellbeing literacy skills, combined with the environmental conditions of a good community, John can realise the capability resulting in wellbeing experiences related to loving connections. The capability model of wellbeing literacy, as per Figure 1, Figure 2 and Figure 3, proposes the development of five components (knowledge, vocabulary, comprehending and composing language, context sensitivity and right intentionality) that interact with internal and external environmental conditions. The capability levels reached depend on the relationship between these factors. If the skill level in the five components is low, or the environmental affordances are low, the capability level is also likely to be lower or nil.

## 5. Contexts for Wellbeing Literacy

Various contexts, such as workplaces, families, schools (especially with the growing focus of wellbeing as the aim of education [62,63,64]), population health and more could be discussed as relevant contexts for application of wellbeing literacy. Here we focus on describing workplaces and families as two examples of wellbeing literacy in these contexts.

### 5.1. Wellbeing Literacy and the Workplace

Wellbeing in the workplace has become of greater interest in recent years [65] and particularly as the impact of the COVID 19 pandemic has been felt around the globe. Restrictions have forced between 30% and 50% of the world’s workforce to work from home [66,67,68] with negative consequences for worker wellbeing including working longer hours with fewer boundaries between ‘work’ and home’, increased levels of technostress, organizational change fatigue and feeling pressure to be constantly online/available [68]. Several aspects of wellbeing literacy make it an appropriate vehicle to enable wellbeing through face-to-face, virtual or hybrid work practices. Firstly, the five components of wellbeing literacy outlined in Table 1 can be present and developed through any working practice format. For example, a team leader with high wellbeing literacy may choose to intentionally create wellbeing experiences for themselves and others by talking with their teams about the concept of antifragility [69,70] and how disruption can be used to grow stronger. This wellbeing literacy practice can be done with all parties face-to-face, with all parties in a virtual environment, or with a mix of some people face-to-face and others joining the conversation via a virtual meeting platform from home. This example also demonstrates the relationships in Figure 3 as the internal environment inputs (the leader’s motivation) interacts with the external environment (the COVID impacted work practices) and with the level of wellbeing literacy (knowledge, skills, composition, comprehending, intention and context) to create wellbeing for the leader and their team. There are numerous further questions to explore including whether workplace interventions are focussed on increasing wellbeing literacy, or whether this varies by profession.

### 5.2. Wellbeing Literacy and Families

Families can provide a natural and rich environment in which to develop and support wellbeing literacy. Gee suggests that primary ‘Discourses’ are developed early in life and within the family through our interactions with parents, siblings and those closest to us [71]. Gee differentiates between discourses (lower case ‘d’), which he describes as “language-in-use” (i.e., language used in context through which activities and identities are enacted), and Discourses (with a capital ‘D’), which are broader than spoken language and include ways of acting, thinking and the values placed on those actions. Discourses both reflect and create the contexts in which they are used. As such, developing wellbeing literacy enables a wellbeing primary discourse that both creates and reflects wellbeing within and between family members, and in the family system as a whole [72].

Using the Capability Model of Wellbeing Literacy, capability to family wellbeing (i.e., a resource that enables wellbeing) could be the capacity for a parent to actively listen. Listening is a generalised ‘wellbeing comprehension’ capability within the Capability Model of Wellbeing Literacy. There is ample evidence that listening builds deep positive relationships in families [53,73,74] and that listening styles such as active, empathic and supportive listening increase positive affect [74,75,76], better coping behaviours, and improved individual and relational health and wellbeing [77,78,79,80]. These are wellbeing ‘achievements or ‘functionings’ for both the individual parent and the family member(s) to whom they are actively listening. This process will be repeated each time the parent engages their active listening capacity to intentionally invest in family relationships and create positive affect for themselves and others in the family context. In doing so, they both create and reflect a wellbeing Discourse within the family.

## 6. Directions for Future Research

The research program of wellbeing literacy has examined both the concept, such as this paper, and how to measure wellbeing literacy [81]. Further important research questions include:How do laypeople define and construct wellbeing through language [8,82,83]?How does wellbeing literacy relate to wellbeing now and over time?How do we increase wellbeing literacy [84]?Is wellbeing literacy a reactive approach, remedial approach, or a preventative approach, or a combination of these?What are the limits of wellbeing literacy?Is wellbeing literacy a mediator and/or moderator of wellbeing interventions [1].

## 7. Conclusions

The capability of language-use (literacy) about and for wellbeing has been introduced as wellbeing literacy. By combining literatures from wellbeing science, literacy, capability approaches, constructivism and contextualism, the multi-components of wellbeing literacy are proposed as the capability model of wellbeing literacy. This concept paper has outlined its formal elements to undergird and complement empirical and practice contributions.

## Figures and Tables

**Figure 1 ijerph-18-00719-f001:**
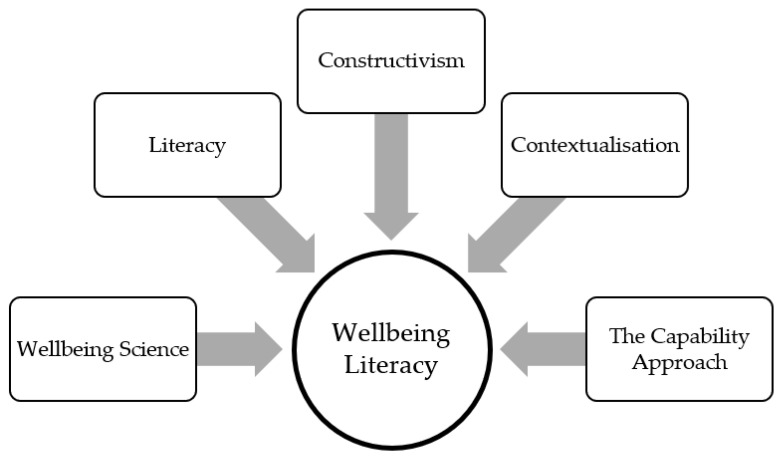
Intellectual and disciplinary contributions to the concept of wellbeing literacy.

**Figure 2 ijerph-18-00719-f002:**
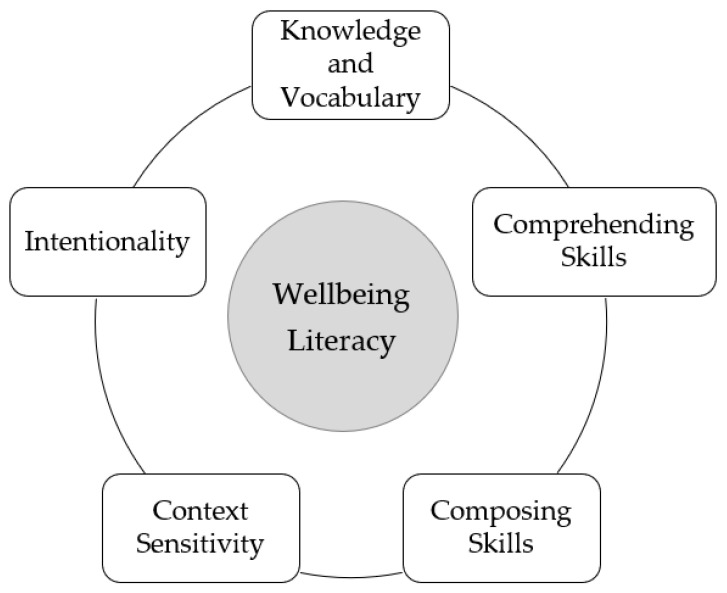
The five components interact to create wellbeing literacy.

**Figure 3 ijerph-18-00719-f003:**
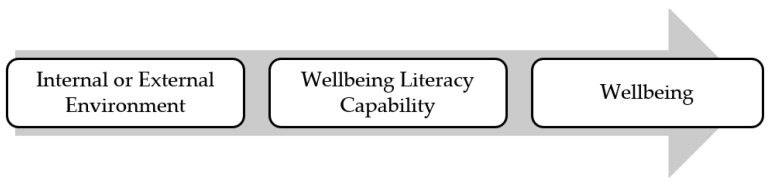
The capability model of wellbeing literacy.

**Table 1 ijerph-18-00719-t001:** Five-component model of wellbeing literacy.

Component	Description
1. Vocabulary and knowledge *about* wellbeing.	Words and basic facts about wellbeing (i.e., content that is signified).
2. Comprehension of multimodal text related to wellbeing.	Reading, listening, viewing about and for wellbeing.
3. Composition of multimodal text related to wellbeing.	Writing, speaking, creating about and for wellbeing.
4. Context awareness and adaptability.	Awareness of differences across contexts and adaptive use of language to fit the relevant context.
5. Intentionality *for* wellbeing.	Habit of intentionally using language to maintain or improve wellbeing of self or others. Includes ethical considerations.

## Data Availability

Not applicable.

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
