# Peer review of "Wellbeing Literacy: A Capability Model for Wellbeing Science and Practice"

_ijerph, 2021, doi:10.3390/ijerph18020719_

Round 1
Reviewer 1 Report
The objective of the manuscript is to propose a conceptual model of well-being literacy, which takes as its central core the capability construct. The authors make a remarkable description of the model and justify its possible contribution to the field of human well-being, both from a theoretical and an applied point of view. While overall this is a well-constructed, grounded, and organized manuscript, there are, in my opinion, some issues that the authors should delve into further. These issues are mentioned below:
- Wellbeing is a difficult construct to define. Even in research within the field of well-being, we can find multiple and heterogeneous ways of conceptualizing this construct. Therefore, the authors should make a brief conceptual approach to this term.
- Lines 126-130: The authors make an interesting reflection on the fact that the educational system tends to prioritize learning related to subject content, relegating training in personal skills to the background. From this consideration, I recommend that the importance of this type of training be deepened, as reflected, for example, by studies focused on the so-called soft skills (e.g., Caggiano, V.; Schleutker, K.; Petrone, L.; González-Bernal, J. Towards identifying the soft skills needed in curricula: Finnish and Italian students’ self-evaluations indicate differences between groups. Sustainability 2020, 12, 4031).
- Well-being literacy and the workplace: This section should be developed in greater depth. Regardless of the situation motivated by COVID, are interventions focused on increasing wellbeing literacy in the workplace? In what kinds of professions? What characteristics do these interventions have? Also, within the relationship between well-being literacy and the workplace, I think it would be interesting to mention the ratio of Losada (see Fredrickson, B.L.; Losada, M.F. Positive Affect and the Complex Dynamics of Human Flourishing. Am. Psychol 2005. 60, 678–686. https://doi.org/10.1037/0003-066X.60.7.678).
- Well-being literacy and families: It would be interesting to analyze the importance of internal dialogue in well-being (for example, self-criticism vs self-compassion).
- Directions for future research: Some interventions in work and family contexts are cited in the manuscript. These are apparently reactive interventions (in problem situations). My question is if preventive interventions are carried out on well-being literacy. If they exist, who are they aimed at (population, ages, ...)? What characteristics do they have? If, on the contrary, these types of interventions are not carried out, do the authors consider that they should be implemented? Please, develop your answer in an argued way. In any case, the practical implications of the proposed model must be analyzed in depth.
Reviewer 2 Report
This is a highly original paper that addresses a neglected aspect of education for wellbeing, namely what the authors call "wellbeing literacy."
Most papers on wellbeing education spend a lot of time discussing the pros and cons of subjective (hedonic) versus objective (eudaimonic) accounts of wellbeing, but this paper can happily avoid that topic, because whichever type of account one favors, wellbeing literacy seems to be an educational/developmental prerequisite of it.
I only have minor comments on the paper and consider it publishable mostly as it is:
1) Since this is a paper broadly situated within the field of wellbeing education, a couple of more citations to the recently burgeoning literatures on wellbeing (esp. flourishing) as the aim of education would be in order. The best overviews are probably in Kristjansson's 2020 Routledge book, Flourishing as the Aim of Education, and in various works co-authored by de Ruyter, see esp. her recent research synthesis: https://oxfordre.com/education/view/10.1093/acrefore/9780190264093.001.0001/acrefore-9780190264093-e-1418
The authors could for example critique briefly the current discourse (de Ruyter, Kristjansson, and also Knoop, H. H., see his (2016). The eudemonics of education. In J. Vittersø (Ed.), Handbook of eudaimonic well-being (pp. 453–471). Dordrecht: Springer) for not paying sufficient attention to the literacy aspect. A brief comment to this effect would make the paper more relevant to students in Education, Educational Philosophy, and Educational Psychology, and more likely to be cited by them.
2) Much has been written lately about "virtue literacy" which is arguably a close cousin of "wellbeing literacy." The authors could cite one or two sources, e.g. https://www.tandfonline.com/doi/abs/10.1080/13617672.2016.1141526
and compare briefly the notions of "virtue literacy" and "wellbeing literacy."
3) The best philosophical account of the value of moral literacy for moral development calls out for at least a quick citation:
Vasalou, S. (2012). Educating virtue as a mastery of language. Journal of Ethics, 16(1), 67–87.
The following paper is also relevant, mutatis mutandis, regarding the historical dimension of wellbeing literacy:
Kesebir, P. & Kesebir, S. (2012). The cultural salience of moral character and virtue declined in twentieth century America. Journal of Positive Psychology, 7(6), 471–480.
